# DIFFERENT RATES FOR DIFFERENT WEIGHTS: DECOUPLED RELATIVE LEARNING RATE SCHEDULES

## ABSTRACT

In this work, we introduce a novel approach for optimizing neural network training by adjusting learning rates across weights of different components in Transformer models. Traditional methods often apply a uniform learning rate across all network layers, potentially overlooking the unique dynamics of each part. Remarkably, our introduced Relative Learning Rate Schedules (RLRS) method accelerates the training process by 13.6%, particularly in complex models such as the Mixture of Experts (MoE). Hyperparameters of RLRS can be efficiently tuned on smaller models and then extrapolated to $27\times$ larger ones. This simple and effective method results in a substantial reduction in training time and computational resources, offering a practical and scalable solution for optimizing large-scale neural networks.

## 1 INTRODUCTION

The learning rate is a crucial hyperparameter in Deep Learning, determining the size of the steps that the optimization algorithm takes when updating model parameters during training. In the context of Transformers, widely used for tasks in Natural Language Processing and other areas, the learning rate significantly impacts the model's convergence and overall performance. While higher learning rates, with larger updates to the model, may generally converge faster, the training also becomes less stable. Therefore, the learning rate must be carefully chosen to balance the speed and stability of the training process.

At the same time, modern Deep Learning architectures are not homogeneous, with different parts having distinct structures, serving varied purposes, and exhibiting unique behaviors. Importantly, they also have individual training dynamics. As seen in Figure 1, gradient updates for different parts follow different paths. This phenomenon also leads to components behaving differently depending on the training phase, which can be problematic in some cases. For example, in Mixture of Experts (MoE) models, the Router often stabilizes early in training, leading to deterministic routing to the Experts (Xue et al., 2024).

Given the diversity of layers within a model, it is reasonable to expect that their requirements would vary, particularly when balancing training speed and stability. Despite this, a uniform learning rate is often applied across all modules. A common practice, for example, is to reduce the learning rate for the whole model after the introduction of an MoE layer due to instabilities (Rajbhandari et al., 2022). As a result, hyperparameters are typically tuned for the entire network, even if, plausibly, the instabilities originate from a single layer. In this work, we relax the implicit assumption of a global learning rate. Since each layer serves a different purpose at different stages of the training process, can we improve it by tailoring the learning rate schedules accordingly?

To answer this question, we decouple learning rates in Transformers and tune them separately for different model components, including Embedding and Unembedding, Attention, Feed-Forward, or, in the case of Mixture of Experts architecture, Router and Experts. By tailoring the learning rate to meet the specific needs of each component, we enhance the model's overall performance and stability.

Furthermore, we propose a simple scheme to adjust relative learning rate values that can be effectively scaled to models larger by orders of magnitude. This approach eliminates the need for extensive hyperparameter searches for larger models, resulting in significant computational savings and enhancing its practical applicability. In essence, we propose the following approach: first, relative LRs should be tuned on a small model; later, the same relative LRs can be reused when training

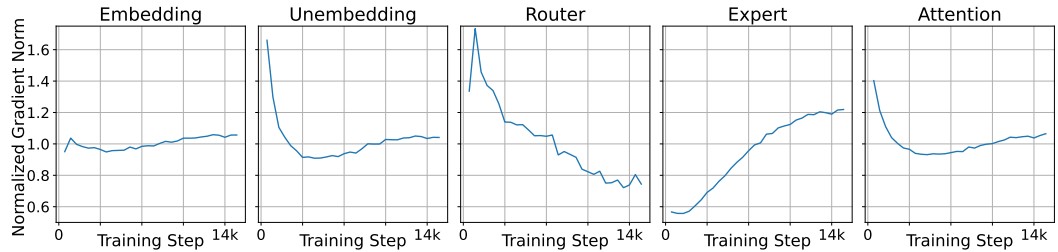

Figure 1: Norms of weight updates after AdamW normalization for different components of the Transformer with MoE.

the model's significantly larger counterpart. Our method is easy to implement, with no additional overhead required, apart from the relatively inexpensive hyperparameter search on the small model. While tailored to our specific training setup, our relative values have proven robust across a range of hyperparameters, making them an excellent starting point. Additionally, we provide an analysis showing how these values, obtained using automated methods, align with our intuitive understanding of Transformer training. In summary,

- We propose distinct, relative learning rate schedules (RLRS) tailored for different components of a Transformer model, optimizing each part individually for better overall performance.
- We show performance improvements of the introduced methods in the standard Transformers with improvements growing in the Mixture of Experts (MoE) based model, highlighting the importance of relative learning rates for more complex models.
- We demonstrate that the hyperparameters tuned on small models extrapolate to larger models, showing that our approach generalizes effectively across different architecture sizes.

## 2 DECOUPLED RELATIVE LEARNING RATE SCHEDULES

We define a *decoupled learning rate* as a separate learning rate schedule for different layer types (also called parts, modules, or components). Decoupled learning rate schedules enable the learning procedure to focus on different components during different phases of a model's pretraining, facilitating more targeted and efficient optimization process.

We specify decoupled learning rates following the structure of the cosine learning rate scheduler (Loshchilov & Hutter, 2016), widely used for training Large Language Models (LLMs) (Touvron et al., 2023; Hoffmann et al., 2024). The cosine scheduler adjusts the learning rate over time according to a cosine function, starting with a high learning rate that gradually decays to a minimum value in a smooth, nonlinear manner.

The parameters we introduce are:

- *Base LR* ($\eta_{base}$) – the reference learning rate for the entire model. In a typical cosine schedule, it is the initial (or maximum) learning rate, $\eta_{start}$, representing the peak value during the training cycle (after a possible period of warm-up).
- *Base LR Final Fraction* ($\lambda_{base}$) – the fraction of the base learning rate that represents the final learning rate at the end of training. The final (or minimum) learning rate is the lowest learning rate value at the end of the training cycle and is given by $\eta_{end} = \lambda_{base} \times \eta_{base}$.

The cosine scheduler adjusts the learning rate following a cosine curve over a specified number of iterations. The learning rate $\eta_t$ at step $t$ is computed using the cosine function, $\eta_t = \eta_{end} + \frac{1}{2}(\eta_{start} - \eta_{end})\left(1 + \cos\left(\frac{t}{T}\pi\right)\right)$, where $t$ is the current step, and $T$ is the total number of steps.

For each component $m$ of the model, we further define a learning rate scaling factor *relative* to the base learning rate $\eta_{base}$:

- *Relative Start LR* ($\lambda_{start}^m$) — the scaling factor of the base learning rate at the beginning of training.
- *Relative End LR* ($\lambda_{end}^m$) — the scaling factor of the final learning rate at the end of training.

Thus, the *decoupled learning rates* $\eta_{start}^m$ and $\eta_{end}^m$ for a component $m$ are defined as:

$$\eta_{start}^m = \eta_{base} \times \lambda_{start}^m \tag{1}$$
$$\eta_{end}^m = \eta_{base} \times \lambda_{base} \times \lambda_{end}^m \tag{2}$$

These values are adjusted for each Transformer component. In this work, we distinguish the following layer modules $m$: Embedding, Attention, Unembedding, and additionally for dense models, the Feed-Forward layer, and for Mixture of Experts models, the Expert and Router layers.

Thus, for a given model, one needs to obtain the baseline learning rate $\eta_{base}$ and the relative learning rates, $\lambda_{end}^m$, $\lambda_{end}^m$. In Section 3.2, we show how we find them in practice. However, in the next section, we demonstrate that we do not need to tune the relative learning rates for every model separately, as the same set of values remains robust across a range of model sizes.

## 2.1 PRESERVING RELATIVE VALUES

In this section, we show that decoupled learning rates remain stable across various models, with their values staying approximately constant relative to the base learning rate.

Tuning relative learning rate values directly on large models may be impractical due to significant computational costs. To address this, we propose a method that fine-tunes these values on a smaller proxy model and then transfers them to a larger model. This approach significantly reduces the need for costly tuning on large models, offering substantial computational savings.

Our method, described in Algorithm 1, involves conducting a search for optimal values on smaller models under the assumption that these relative values extrapolate effectively to larger models. This search consumes only a fraction of the training time required for large models.

---

**Algorithm 1** Relative LR Adjustment Algorithm

1: Find $\eta_{base}$ for a small model.
2: For each module $m$, find relative values $\lambda_{start}^m$ and $\lambda_{end}^m$ on a small model.
3: Find base learning rate $\eta_{base}$ for the large model.
4: Apply relative learning rates $\lambda_{start}^m$ and $\lambda_{end}^m$ from the small model.

---

While we do not claim that $\lambda_{start}^m$ and $\lambda_{end}^m$ values are optimal for larger models, they are easy to use and yield substantial improvements, as shown in the next section. We leave the investigation of optimal extrapolation as future work.

Algorithm 1 presents a simple way to find relative learning rates independently of the base learning rate and apply them to both the small model and the large model. This is particularly desirable if we are given a model with a base learning rate that has already been tuned. However, if this is not the case, alternatively to Algorithm 1, we propose Algorithm 2, where the base learning rate is adjusted again after the relative learning rates.

---

**Algorithm 2** Relative LR Adjustment Algorithm

1: For a small model, find $\eta_{base}$ and for each module $m$, find relative values $\lambda_{start}^m$ and $\lambda_{end}^m$.
2: Apply relative learning rates $\lambda_{start}^m$ and $\lambda_{end}^m$ from the small model.
3: Adjust base learning rate $\eta_{base}$ for the large model.

---

Algorithm 2 may perform slightly better than Algorithm 1. However, for practical purposes, applying relative rates to an already tuned base model offers substantial gains, and we focus on this setting in the next section. Additionally, the implementation of Step 1 of Algorithm 2 can be further expanded. We provide the details of our implementation in Section 3.2 and Algorithm 3.

## 3 RESULTS

### 3.1 EXPERIMENTAL SETUP

All models used in this study are decoder-only Transformers trained on the C4 dataset (Raffel et al., 2019). The GPT-2 tokenizer (Radford et al., 2018) is employed. We optimize using AdamW (Loshchilov & Hutter, 2019) and apply cosine decay with a linear warmup for the first $1\%$ of training steps. For better stability, weight initialization follows a truncated normal distribution with a reduced scale, as suggested by Fedus et al. (2022). Mixed precision training is used, with Attention and Router calculated at high precision. The models use SwiGLU activation and Token Choice routing with 8 Experts, of which 1 is activated. We use two auxiliary losses for the Router: z-loss with a weight of $0.001$ (Zoph et al., 2022) and load balancing with a weight of $0.01$ (Fedus et al., 2022). Compute-optimal training durations are based on Hoffmann et al. (2024), calculated for MoE as $20\times$ the number of active parameters excluding Embedding and Unembedding, as recommended in Ludziejewski et al. (2024). Moreover, we provide one comparison on overtrained $\text{MoE}_{8\times113\text{M}}$, with almost 130 token to active parameter ratio. For all extrapolations, we tune base learning rates separately for RLRS and the baseline with the precision of a grid defined by $\{1e{-}n, 2e{-}n, 5e{-}n\}$.

For both dense and MoE models, the weight decay value has been optimized to $0.1$, the initialization scale to $0.15$, and *Base LR Final Fraction ($\lambda_{base}$)* to $0.04$ for MoE and $0.06$ for dense.

| Type | Active Params | Total Params | $d_{model}$ | $n_{layers}$ | $n_{experts}$ | BS | SL |
|---|---|---|---|---|---|---|---|
| $\text{MoE}_{8\times34\text{M}}$ | 33.6M | 210M | 512 | 8 | 8 | 256 | 512 |
| $\text{Dense}_{34\text{M}}$ | 33.6M | 33.6M | 512 | 8 | 8 | 256 | 512 |
| $\text{MoE}_{8\times113\text{M}}$ | 113M | 708M | 768 | 12 | 8 | 256 | 512 |
| $\text{Dense}_{113\text{M}}$ | 113M | 113M | 768 | 12 | 8 | 256 | 512 |
| $\text{MoE}_{8\times906\text{M}}$ | 906M | 5.67B | 1536 | 24 | 8 | 384 | 1024 |
| $\text{Dense}_{906\text{M}}$ | 906M | 906M | 1536 | 24 | 8 | 384 | 1024 |

Table 1: Models used in this paper. BS indicates batch size, and SL indicates sequence length.

In Tables 2 and 3, we report a speedup metric that measures how much faster a training process becomes when relative rates are applied. It is calculated using $(\frac{T_{\text{base}}}{T_{\text{relative}}} - 1) \times 100\%$, where $T_{\text{base}}$ is the number of steps performed in the standard training with a base learning rate, and $T_{\text{relative}}$ is the number of steps incurred until the loss of the training with the relative learning rate schedule exceeds baseline loss. It is important to note that using this metric likely underestimates the improvement of our method since for relative learning rate training steps, when we compute the speedup, the cosine schedule has not yet reached its end. We perform three runs for each configuration, except for $\text{Dense}_{906\text{M}}$, due to compute limitations. For each run, we measure the loss per $S$ steps, where $S$ is $1\%$ of all training steps. The speedup is calculated over the means of 3 runs. To reduce variance from random data seeds, we use 3 specified data seeds for each model type comparison.

### 3.2 FINDING DECOUPLED RELATIVE LEARNING RATES

Our method involves determining a set of relative learning rates. While these hyperparameters could be optimized using a straightforward grid search, such a procedure requires carefully setting the search boundaries and involves an exponential number of training runs. In our experiments, we opt for a more scalable local search algorithm, which is described below.

---

**Algorithm 3** Local Search

---

1: Iterate over the set of hyperparameters.
2: For a given hyperparameter, multiply its value by a factor from $\left\{\frac{1}{5}, \frac{2}{3}, \frac{3}{2}, \frac{5}{1}\right\}$
3: Run experiments, and if there is an improvement, adjust the hyperparameter value.
4: If any change has been made among all hyperparameters, return to Step 1.

---

To ensure proper configuration, we optimized the weight decay and initialization scale along with all RLRS values. For the baseline, the same algorithm was used to find the learning rate at the start and at the end of the cosine schedule along with weight decay and initialization scale.

## 3.3 TUNING SMALL MODELS

A small model allows for a wider search of hyperparameters. Local search runs around 500 experiments on $MoE_{8\times34M}$ models before converging and 200 on $Dense_{34M}$.

Adjusting relative rates according to the proposed schedule results in substantial gains in the form of shortening the training by up to 23% for MoE and by up to 17% for a dense Transformer (see Table 2).

| Type | LR Type | Base LR | Train Tokens | Speed-Up |
|---|---|---|---|---|
| $MoE_{8\times34M}$ | baseline | $3 \times 10^{-3}$ | 1.3B | - |
| | relative | $3 \times 10^{-3}$ | 1.3B | 22.8% |
| $Dense_{34M}$ | baseline | $2 \times 10^{-3}$ | 1.3B | - |
| | relative | $2 \times 10^{-3}$ | 1.3B | 17.2% |

Table 2: Using RLRS results in faster model convergence.

## 3.4 EXTRAPOLATION

In this section, we show the results when relative values tuned on a small model are extrapolated to larger models. We use models with 113M and 906M active parameters (in the case of MoE, 707M and 5.7B total parameters respectively). As shown in Table 3, extrapolating the relative rates results in up to 13.6% faster training in case of MoE and 7.7% in case of dense model. While in both cases gain was higher on our small proxy model, it is reasonable to assume that some of it was due to overfitting to the specific setting.

In Figure 4, we show that without fine-tuning on a large model, the transferred relative values are still noticeably better than the baseline, and generally close to the optimal value. The only exception is the Embedding layer. We elaborate on these findings in the next section.

| Type | LR Type | Base LR | Train Tokens | Speed-Up |
|---|---|---|---|---|
| $Dense_{113M}$ | baseline | $1 \times 10^{-3}$ | 2.5B | - |
| | relative | $1 \times 10^{-3}$ | 2.5B | 17.5% |
| $MoE_{8\times113M}$ | baseline | $2 \times 10^{-3}$ | 2.5B | - |
| | relative | $1 \times 10^{-3}$ | 2.5B | 19.0% |
| $MoE_{8\times113M}$ (overtrained) | baseline | $1 \times 10^{-3}$ | 14B | - |
| | relative | $1 \times 10^{-3}$ | 14B | 14.6% |
| $Dense_{906M}$ | baseline | $5 \times 10^{-4}$ | 20B | - |
| | relative | $5 \times 10^{-4}$ | 20B | 7.7% |
| $MoE_{8\times906M}$ | baseline | $2 \times 10^{-4}$ | 20B | - |
| | relative | $2 \times 10^{-4}$ | 20B | 13.6% |

Table 3: Gains from extrapolating relative learning rates to larger models.

# 4 ANALYSIS

## 4.1 INTERPRETING RELATIVE LEARNING RATES

In this section, we present the numerical results and trends for relative learning rates (RLRS) and analyze them with respect to each layer module. We prioritize MoE models, as RLRS yields more substantial gains in this setting. Although the values have been determined experimentally, they are often interpretable and aligned to counteract the existing issues of each component.

**Embedding.** The relative learning rate $\lambda_{start}$ starts high (3.3 for MoE and 5 for dense) and decays to 0.6. This aggressive early training helps the Embedding stabilize quickly, as it influences the entire network. Later in the training process, the learning rate is reduced to prevent drastic changes in the Embeddings, ensuring the rest of the model can adjust accordingly. As seen in Section 4.3, this is the only layer that prefers adjustment of relative learning rate when increasing model size; that is, while other relative learning rates transfer without change, the Embedding's rate should also be increased.

**Unembedding.** Unembedding handles the conversion of the model output into a probability distribution over the tokens in its vocabulary. We observe that, similar to the Embedding, the relative learning rate gradually decreases toward the end of training. This behavior aligns with observations in the literature that weights in the Unembedding may diverge, potentially causing instabilities later in the training process (Chowdhery et al., 2022; Zoph et al., 2022), which would require reducing gradient values.

**Router.** Router (or gating network) plays a crucial role in determining which Expert networks are trained during the learning process. However, it has been observed that the model often learns its routing decisions early in the pre-training phase, and these decisions remain largely fixed throughout training (Xue et al., 2024). Once a token is assigned to an Expert, it is rarely reassigned, making it difficult for the model to adapt to new or unseen data during later stages of training. Moreover, the Router tends to be unstable at the early stages of training. Starting the relative learning rate with a lower value of 0.6 and then ending with 1 might help mitigate these problems.

**Experts.** The relative learning rate of an Expert layer increases from the smallest value of 0.3 to aid stability when the Router is essentially random, and increases to the highest relative value at the end, allowing the Experts to fine-tune while the Router remains largely fixed.

**Attention.** In Attention projection layers, the learning rate remains unchanged in the MoE model, making it unique in not benefiting from relative rates.

We summarize the Decoupled Learning Rate for both dense and MoE models in Table 4.

|  | Embedding | Unembedding | Router | Experts | Attention |
|---|---|---|---|---|---|
| start | 5.0 | 0.6 | 0.6 | 0.3 | 1 |
| end | 0.6 | 0.4 | 1 | 1.125 | 1 |

Table 4: Relative learning rate values ($\lambda$) for MoE.

|  | Embedding | Unembedding | Feed-Forward | Attention |
|---|---|---|---|---|
| start | 5 | 1 | 1 | 1 |
| end | 0.6 | 0.4 | 0.6 | 0.2 |

Table 5: Relative learning rate values ($\lambda$) for dense models.

## 4.2 STABILITY

In this work, we demonstrate that applying relative learning rates enhances stability across models of varying sizes. Training Transformers, particularly MoE architectures, often leads to instabilities when using a uniform learning rate across the entire architecture. As seen in Figure 2, the baseline exhibits loss spikes that were absent with the relative schedules. This is also intuitive, as MoE models are considered unstable and require lower learning rates for optimal learning, which, however, affects the speed of training. In our method, the learning rates for both the Router and the Experts start off relatively lower, while they are higher for other parts of the model, resulting in both better stability and convergence.

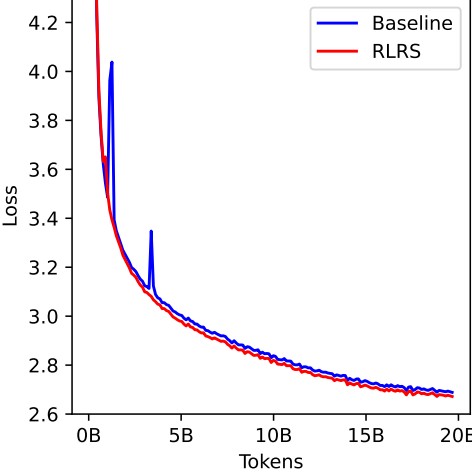

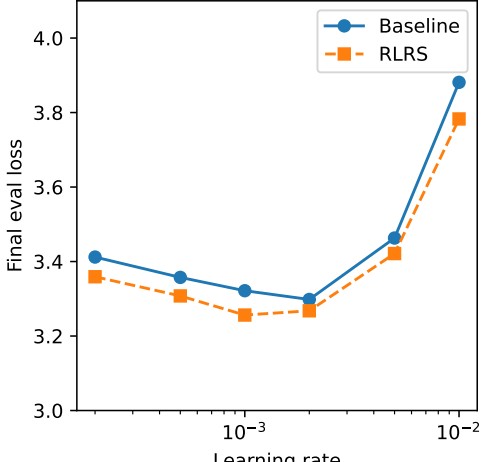

Figure 2: Stabilizing training with RLRS vs. baseline LR for large Mixture of Experts (906M).

Figure 3: Loss of RLRS and baseline for different $\eta_{base}$ on $\text{MoE}_{8\times113\text{M}}$ models. RLRS results in better loss across a range of learning rates.

We further argue that training a model with relative learning rates enhances stability across different baseline learning rates, reducing sensitivity to parameter tuning and improving training stability, especially in large models. Large Transformer-based models frequently encounter instabilities, even when using hyperparameters that worked well for smaller models. Wortsman et al. (2023) demonstrate that instabilities in small models with a higher than optimal learning rate can be a good proxy measure for instabilities on a larger scale. Following that, we provide Figure 3 comparing the learning rate sensitivity of RLRS and the baseline. We can see that training with relative learning rates outperforms the baseline across various learning rates.

## 4.3 RELATIVE LEARNING RATES FIT FOR LARGE MODELS

In Figure 4, we show that the transferred rates $\lambda_{start}^m$ and $\lambda_{end}^m$ perform consistently well compared to the surrounding values. This empirical result shows the relative rates not only result in gains as shown in Section 3.4, but also are close to optimal for other models. An interesting exception is the Embedding layer, which shows a clear preference to increase its relative learning rate when increasing the model size. This aligns with Lingle (2024), which studies models with increasing width and finds that Embedding is the only layer type whose learning rate should not be scaled down when increasing the model's layers.

## 4.4 ABLATIONS

Figure 4 also shows the importance of tuning the relative learning rates for individual modules. The study indicates the particular importance of Embedding and Unembedding. It is important to note that the improvement brought by the method comes largely from the interactions between the relative rates for all the components, rather than any specific module.

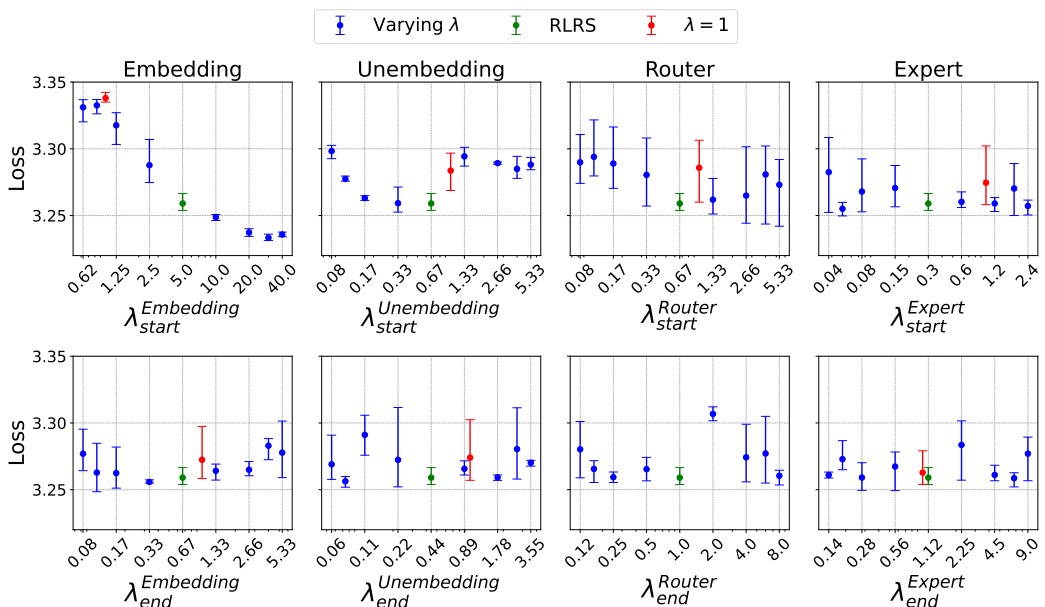

Figure 4: The performance of $\lambda_{\text{small}}$ extrapolated to large models. The optimal $\lambda_{\text{small}}$ (green) is compared to larger and smaller relative rates (blue) and the baseline learning rate (red). In most cases, the $\lambda_{\text{small}}$ relative learning rates are close to optimal values in larger models. The Embedding layer requires scaling relative values.

## 5 RELATED AND FUTURE WORK

The literature on learning rates in Machine Learning, particularly for Transformers, highlights the importance of adaptive learning rate schedules. Stochastic Weight Averaging (SWA) (Izmailov et al., 2018) utilizes a modified learning rate schedule that applies a decaying learning rate during the initial phase of training, followed by a constant rate for the remainder. In Sun et al. (2019), the authors introduce layer-wise learning rate decay, which applies higher learning rates to top layers and lower rates to bottom layers. A related concept, discriminative fine-tuning, is discussed in Howard & Ruder (2018). Additionally, Everett et al. (2024) explores how various parameterizations and optimizers impact the learning process in large-scale models and proposes a per-layer learning rate strategy.

### 5.1 COMBINATION WITH TENSOR PROGRAMS

Our method explores the transfer of relative learning rates; however, the base learning rate must still be independently tuned for the extrapolated model. Approaches such as Tensor Programs (Yang, 2020; Yang et al., 2022) propose parameterizations that facilitate the transfer of the base learning rate. By combining these two approaches, it may be possible to achieve a zero-shot transfer of RLRS.

While our methods share similarities with Tensor Programs and draw inspiration from them, our project has a distinct goal. We aim to identify implicit assumptions in the tuning process and decouple parameters to devise a scheme that enables Large Language Models (LLMs) to converge in fewer steps. Our extrapolations demonstrate that our optimization scheme depends on the architecture rather than the model size. This scheme is defined relative to the base learning rate, which must be tuned individually for each model size. Our method does not aim to facilitate learning rate transfer between different model sizes and is supported by experimental evidence. We do not mathematically examine the limits of parameter updates in a gradient descent step. A key difference is that our relative values change dynamically during training, and our goal is to enable the model to focus on different aspects during pretraining.

## 5.2 FINE-TUNING

Fine-tuning allows users to adapt pre-trained Large Language Models (LLMs) to more specialized tasks. In traditional fine-tuning, certain model components are often "frozen" (effectively setting their relative learning rates to zero) to preserve learned knowledge while adapting other parts. Our proposed method introduces a more flexible approach, serving as a continuous alternative to freezing parameters. This enables fine-grained control over information transfer within specific components of the model. Consequently, our method could be particularly applicable to fine-tuning scenarios and complement existing methods that involve freezing parameters. Parameter-Efficient Fine-Tuning (PEFT) techniques, such as LoRA (Hu et al., 2021), address this by updating only a subset of parameters while freezing the rest. Our work aligns with more advanced methods like LoRA+ (Hayou et al., 2024), which select different learning rates for the adapter matrices, and AdaLoRA (Zhang et al., 2023), which adapts the rank of the LoRA matrices, providing enhanced flexibility in the fine-tuning process.

## 6 CONCLUSION

We have presented a method for decoupling learning rate schedules across different neural network components, removing the implicit assumption of homogeneity among them, and achieving better training speed and stability as a result. This method applies to any Transformer-based model and significantly enhances performance in Mixture of Experts (MoE) models. By tuning relative learning rates on smaller models, this approach can be used to economically achieve significant improvements in the training of order-of-magnitude larger models.

## REPRODUCIBILITY

Our method is straightforward to implement and clearly outlined in Section 2, making it easy for others to replicate. For the camera-ready version, we will share the full code and configuration files used in our experiments through a public repository, as the current code is hard to anonymize properly. All hyperparameters are documented in detail within the main text.

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

## A  ADDITIONAL FIGURES

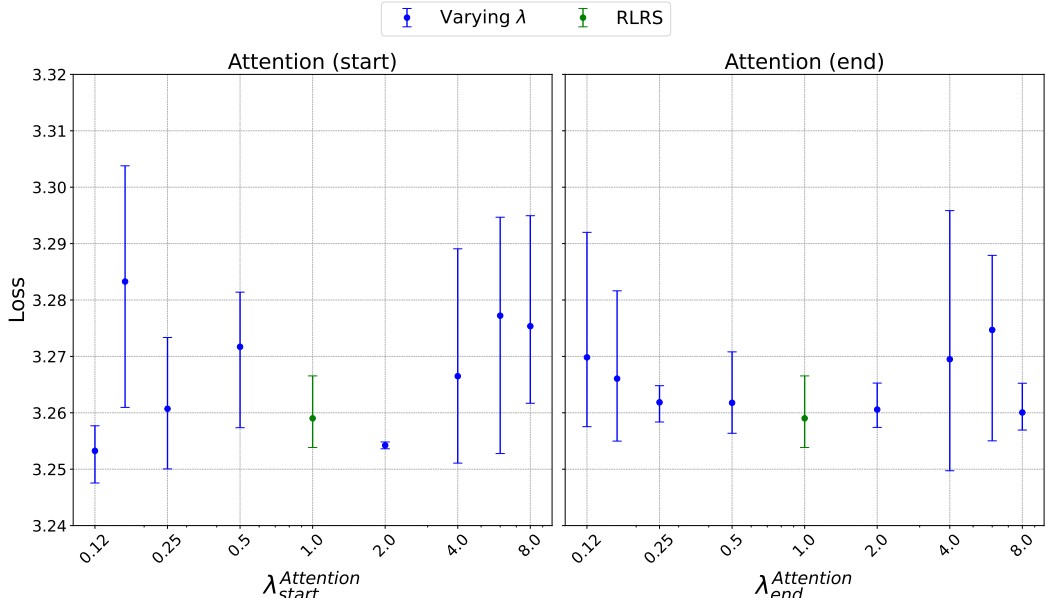

Figure 5: Varying $\lambda_{start}^{Attention}$ and $\lambda_{end}^{Attention}$ for MoE$_{8 \times 116M}$. For this component, our optimization algorithm kept the relative value unchanged ($\lambda = 1.0$).

