# OpenReview forum: "Different Rates for Different Weights: Decoupled Relative Learning Rate Schedules"
_ICLR.cc/2025/Conference — ICLR 2025 Conference Withdrawn Submission_

### Official Review · Reviewer_V9H1 · 2024-10-19

**Soundness:** 1
**Presentation:** 2
**Contribution:** 1
**Rating:** 1
**Confidence:** 5

**Summary:**

This paper proposes to have different learning rates for different parts of the transformer model. The authors propose a simple heuristic for deciding what the best learning rate is for each component. Their method, RLRS, improves training speed and stability. The proposed approach shows a 13.6% acceleration in training time and can scale effectively to models up to 27 times larger without the need for extensive re-tuning of hyperparameters.

**Strengths:**

• I think the topic the authors are aiming to address is interesting, and it makes sense that different learning rates for different components would help to increase convergence and performance.

**Weaknesses:**

• This proposed method is not theoretically motivated and the proposed method appears to be a very crude heuristic for selecting the learning rate. Algorithm 3, which discusses how the hyperparameters are found suggests that the authors are just doing a kind of random search, where you multiply the values by a factor from (x0.2 - x5).

• There is very little background or discussion covering the central topics of this paper. For example, there is almost no time spent discussing contemporary hyper-parameter optimization techniques. Furthermore, the idea of giving different components different learning rates is not a new one – in fact it is natively supported in libraries such as PyTorch.

**Questions:**

N/A

---

### Official Review · Reviewer_T5Xq · 2024-11-01

**Soundness:** 2
**Presentation:** 3
**Contribution:** 1
**Rating:** 3
**Confidence:** 3

**Summary:**

This paper proposes using different peak learning and final learning rates for different types of layers in GPT training across both dense and mixture-of-expert configurations. They show that the adjustments can be tuned on a small scale and transferred to larger models, providing a speedup (reduced tokens for a given loss) on the order of 10-20% in both cases. The paper then analyzes and interprets some of the tuned adjustment values.

**Strengths:**

- Problem tackled is practical and of interest to the community.
- The core idea is interesting.
- The paper is clear and easy to follow.
- The authors tune their hyperparameters well.

**Weaknesses:**

- The overall novelty of the work might be limited given that tuning learning rates for each component of the network is already well known e.g. in muP works [1] [2].
- The paper doesn’t answer the question of whether the final lr values are truly needed. Figure 4 suggests that training is mostly sensitive to the peak lr of different components, but this is exactly what existing work does.
- The experiments are relatively small in scale and could be better ablated.

[1]: https://arxiv.org/abs/2407.05872
[2]: https://arxiv.org/abs/2407.17465

**Questions:**

Recommendation: Overall I recommend rejection based on seemingly limited novelty over existing approaches combined with the lack of other significant contributions (e.g. a large-scale validation or a theoretical analysis of these approaches could be valuable despite the method not being novel).

Suggestions:
- I recommend trying to add stronger evidence that the final learning rate values are needed to justify your approach over existing work.
- I somewhat doubt that schedules for short runs transfer well to longer runs. For example weight decay can be seen as an (effective) learning rate scheduler but it only makes a significant difference for longer runs. It might be more convincing if you could provide an example or arguments to support this approach.
- In Figure 1 you show the update size over time and use this to justify the learning rate schedules. Some other optimizers like Lion would fix the update size. Would you expect different schedules for different parameters to help in this case?
- If you pursue using schedules for each parameter it would be nice to see a greater difference, e.g. something like different warmup lengths or earlier decay with a WSD. This would be a greater differentiator from the peak lr.

---

### Official Review · Reviewer_ETGF · 2024-11-01

**Soundness:** 2
**Presentation:** 2
**Contribution:** 1
**Rating:** 3
**Confidence:** 2

**Summary:**

This paper proposes Decoupled Relative Learning Rate Schedules (RLRS) for Transformers, assigning distinct learning rates to different components like Embedding and Attention layers. The authors suggest tuning these rates on smaller models and then applying them to larger ones to save on computational costs. Experiments with dense and Mixture of Experts (MoE) models show improved training speed and stability, particularly in large-scale MoE setups. The results indicate that RLRS can scale efficiently, offering an approach to optimize Transformer training at larger scales.

**Strengths:**

1. The proposed method has been applied to large model scales, which is practical if the implementation and results can be verified.

**Weaknesses:**

1. (**Limited Novelty of proposed approach**) The authors assume that relative learning rates tuned on smaller models will perform equally well on much larger models. A similar idea has been proposed in [1], but there is no discussion of comparing it with related work.

2. The rationale for introducing separate learning rate schedules for different components is not well-explained. The rationale and motivation mentioned in the paper are only listed below, which is not convincing.
> At the same time, modern Deep Learning architectures are not homogeneous, with different parts on the training phase, which can be problematic in some cases. For example, in Mixture of Experts (MoE) models, the Router often stabilizes early in training, leading to deterministic routing to the Experts

3. The paper does not sufficiently compare its approach with established methods for adaptive learning rates, such as [2].


**Reference**
[1] Tensor Programs V: Tuning Large Neural Networks via Zero-Shot Hyperparameter Transfer

[2] Sophia: A Scalable Stochastic Second-order Optimizer for Language Model Pre-training

**Questions:**

1. could the author provide the training curves or the measurement of training time to verify the time reduction reported in the paper?
2. What is the computational overhead associated with tuning the relative learning rates on smaller models? A breakdown of the resources required to tune these rates on smaller models and transfer them to larger ones would clarify the practical efficiency of this approach in real-world applications.

---

### Official Review · Reviewer_ciJG · 2024-11-04

**Soundness:** 3
**Presentation:** 2
**Contribution:** 1
**Rating:** 3
**Confidence:** 3

**Summary:**

Conventionally, when training  a transformer model, the learning rates and the learning rates schedules of parameters in the entire model are set the same. However, recent works have shown that the weight updates of different modules in a Transformer model have varied dynamics, indicating that letting all modules share the same learning rate schedule may decrease the stability of the training process. Motivated by this, this paper has proposed a module-adaptive learning rate schedule called Relative Learning Rate Schedules (RLRS), which fine-tune the learning rate schedule for each module adaptively using a proposed approach called Local Search. The paper has also suggested that the RLRS's of a smaller model can be preserved to be applied by a larger model without much loss of performance. Experiments have been conducted to show the efficiency and effectiveness of the proposed RLRS.

**Strengths:**

1. Has a valid and strong motivation.
2. The finding that the RLRS's of a smaller model can be transfered (with slight adjustment) to a larger model without losing much fitness on the larger model is intriguing and practically meaningful.
3. The experiments are comprehensive and do show the effectiveness and efficiency of the proposed RLRS.

**Weaknesses:**

1. The validity of the claim that the hyperparameters of small model remains robust in large model is not convincingly demonstrated.
2. The hyperparameter tuning process is based on plain searching, which make the algorithm lack of efficiency in practice. Also, in this context, it's not convincing enough to say that the improvement of RLRS comes from using different learning rates schedules for different modules. It's simply because we have introduced more hyperparameters so that we have more degree of freedom to fine-tune the training process, just like in a standard deep learning task, if the model has more parameter, it's natural to predict that the training performance of the model will be much improved.

**Questions:**

1. Line 123 to 124 on page 3. Here the authors indicate that they demonstrate that the same set of relative learning rates remains robust across a range of model sizes "in the next section". However, in the next section 2.1, I don't see any demonstrations of the claim, but only introductions of the relative LR adjustment algorithms. Also, although Figure 4 does somehow show that the performance of $\lambda_{small}$ extrapolated to large models remains optimal or nearly optimal, it still hasn't demonstrated the "across 'a range of model sizes'" part in the claim.
2. Algorithm 1 and Algorithm 2 are named the same: "Relative LR Adjustment Algorithm". Although they can be distinguished by simply indicating algorithms 1 or algorithm 2, it's still better to give them distinct names since they seem to be an essential part of the paper.
3. Line 159. What "substantial gains" exactly has applying relative rate to an already tuned base model offered?
4. Algorithm 3, step 2. How are the set of the 4 values derived?
5. Algorithm 3, step 3. How is an "improvement" defined? Also, speak of improvement, I believe we are comparing the current experiment with the previous one. Then in this context, how is the very first experiment set up and run?
6. Section 3.3. This section indicates that after we have got the best learning rate schedules for both RLRS model and baseline model, the former one can be trained faster than the latter one. But is the process of fine-tunning taken into account? Meaning, will the process of fine-tunning in RLRS take much longer time that that in baseline, since there are more hyperparameters to be tuned, and the process seems to be based on plain searching?
7. Should we care about the changes in the test loss between RLRS trained model and baseline trained model?
8. Section 4.1. Is the analysis in this section conducted on a small model or a large one?
9. Section 4.1, Attention. Is there an explanation or an insight behind the behavior of the Attention module that the learning rate remains unchanged? Also, the authors haven't analyzed the behavior of learning rates in the Attention module in the dense models, which is illustrated in Figure 5.
10. Table 4. The start value of Embedding should be 3.3 right, according to line 279 to 280?
11. Whenever mentioning "performance" or "loss" etc., please specify which kind of performance or loss (training or test), e.g., Figure 2, and other places.
12. Does the stability of the training process necessarily lead to better eval loss?
13. Line 375. The authors claim the Figure 4 also shows the importance of tuning the relative learning rate for individual modules. Please explain this claim from the plots in Figure 4.

---

### Note · Authors · 2024-12-03

**Comment:**

**Summary**

We thank all the reviewers for their thorough feedback and valuable insights. Based on their comments, we recognize that our paper requires substantial revisions, particularly in terms of writing. Therefore, we have decided to withdraw the submission. We understand the concerns raised and provide detailed responses to reviewers below, outlining our planned improvements. We acknowledge that many significant issues mentioned in the reviews stemmed from a lack of adequate explanations and comparisons in our paper. We apologize for the rushed work and aim to do better in the future. We are grateful for all the feedback, as it will significantly help us improve the paper. We maintain our belief in the presented research work while working on further revision.

**Regarding strengths**

We sincerely appreciate the positive feedback and constructive insights provided by the reviewers. We are pleased that Rev. ciJG recognizes our "valid and strong motivation" and finds our discovery that "the RLRS's of a smaller model can be transferred (with slight adjustment) to a larger model without losing much fitness" both "intriguing and practically meaningful." We also value the acknowledgment of our "comprehensive experiments" that demonstrate the "effectiveness and efficiency" of our proposed RLRS. Rev. ETGF highlights the practical application of our method "to large model scales," reinforcing its utility and relevance. Rev. T5Xq emphasizes the relevance of the problem we address and proper hyperparameter tuning. Similarly, Rev. V9H1 supports the "interesting" topic and the logic behind our approach, emphasizing that varying learning rates "makes sense" for improving convergence and performance. We are encouraged by these endorsements and remain committed to advancing this line of research.

**Regarding Tensor Programs V**

Some reviewers suggested that no comparison is made between our work and Tensor Programs. However, such a comparison can be found in Section 5.1 (“Combination with Tensor Programs”). While at first glance our work might seem similar to muP / Tensor Programs, we believe there are some crucial differences. Firstly, we do not aim to transfer learning rate from small to large models - while differently-sized models can have different optimal learning rates, our relative learning rates (the coefficients) can be transferred from small to large models while preserving a large part of the improvements. Furthermore, we believe Tensor Programs and RLRS could be unified, as they serve different purposes, enabling both faster convergence and a fuller transfer of hyperparameters.

**Regarding adaptive optimizers like Sophia**

One reviewer suggested comparing our method to adaptive normalization, like Sophia, which dynamically adjusts the learning rate during training based on feedback from gradients or other training dynamics. However, our approach takes a different direction: instead of dynamically adjusting learning rates based on gradient behavior, we use fixed relative scaling factors for learning rates across different components, tuned on a smaller proxy model. Moreover, our method is designed to work with any underlying optimization algorithm. Unlike adaptive methods that strive to find better minima, our goal is to improve training efficiency and performance specifically for Transformers and Mixture of Experts (MoE) models. As such, the proposed method is orthogonal to the existing literature on adaptive learning rates.

**Regarding the local search algorithm**

Reviewers have highlighted a few issues with the local search algorithm used to tune the relative learning rates, RLRS, such as its inefficiency due to being a simple search algorithm and the fact that it has its own hyperparameters. At the same time, the reviewers perceive the lack of comparison of our approach with other contemporary hyperparameter search methods as a weakness of our work. The local search algorithm we employ is not our main contribution and can easily be replaced by another technique. It is simply an example of an algorithm that could be used to find the relative learning rates. We will ensure to make that clearer in future versions of our manuscript.

Furthermore, one reviewer posits that simply introducing more hyperparameters will trivially allow us to improve the quality of the training procedure. While this critique could be applied if we did not extrapolate our experiments, we argue that it is not valid in our case. We transfer the relative learning rates fitted on small experiments to much bigger models, trained on many more tokens, preserving the speed-up to a large extent. Moreover, we show that our technique interacts with other hyperparameters in a non-trivial manner by showing it improves the final loss across a range of learning rates (Figure 3), while also enhancing the training stability (Figure 2).

**Regarding the motivation and the rationale**

We appreciate the feedback regarding the need for clearer motivation and rationale. Our work addresses the critical role of the learning rate in Transformer models by decoupling and tuning relative learning rate schedules for different model components. Furthermore, we propose a scalable approach where relative learning rates, tuned on small models, can be effectively applied to much larger models. This eliminates the need for extensive hyperparameter searches and results in significant computational savings. We will strive to improve the clarity of our writing and better articulate the motivation behind our approach in the next version of the paper.

**Regarding the scale of experiments**

Some reviewers suggested that the claim about the transferability of the relative learning rates (RLRS) to larger models is not convincingly demonstrated. While it is true that many techniques require validation at a proper scale, we believe that we have sufficiently demonstrated our findings. We transfer the RLRS from models with 210 million parameters to models with 5.67 billion parameters (a 27x increase), and from training runs on 1.3 billion tokens to runs on 20 billion tokens (a 15x increase). This constitutes over 400x increase in training FLOPs, which we consider both convincing and challenging to exceed on an academic budget. We believe that placing greater emphasis on the scale increase between the source and target training runs of the transfer could make our claims more convincing. We acknowledge that we have not made this point clear and will address this issue in future versions of the manuscript.

**Withdrawal Confirmation:**

I have read and agree with the venue's withdrawal policy on behalf of myself and my co-authors.